# Cost-effectiveness of pessary therapy versus surgery for symptomatic pelvic organ prolapse: an economic evaluation alongside a randomised non-inferiority controlled trial

Ângela J Ben ![ORCID],[1] Lisa R van der Vaart,[2] Judith E. Bosmans,[3] Jan-Paul W R Roovers,[4,5] Antoinette L M Lagro-Janssen,[6] Carl H van der Vaart,[7,8] Astrid Vollebregt,[9] PEOPLE group

For numbered affiliations see end of article.

**Correspondence to**
Dr Ângela J Ben;
a.jornadaben@vu.nl

## ABSTRACT

**Objective** To evaluate the cost-effectiveness of pessary therapy as an initial treatment option compared with surgery for moderate to severe pelvic organ prolapse (POP) symptoms in secondary care from a healthcare and a societal perspective.

**Design** Economic evaluation alongside a multicentre randomised controlled non-inferiority trial with a 24-month follow-up.

**Setting** 21 hospitals in the Netherlands, recruitment conducted between 2015 and 2022.

**Participants** 1605 women referred to secondary care with symptomatic prolapse stage ≥2 were requested to participate. Of them, 440 women gave informed consent and were randomised to pessary therapy (n=218) or to surgery (n=222) in a 1:1 ratio stratified by hospital.

**Interventions** Pessary therapy and surgery.

**Primary and secondary outcome measures** The Patient Global Impression of Improvement (PGI-I), a 7-point scale dichotomised into successful versus unsuccessful, with a non-inferiority margin of −10%; quality-adjusted life-years (QALYs) measured by the EQ-5D-3L; healthcare and societal costs were based on medical records and the institute for Medical Technology Assessment questionnaires.

**Results** For the PGI-I, the mean difference between pessary therapy and surgery was −0.05 (95% CI −0.14; 0.03) and −0.03 (95% CI −0.07; 0.002) for QALYs. In total, 54.1% women randomised to pessary therapy crossed over to surgery, and 3.6% underwent recurrent surgery. Healthcare and societal costs were significantly lower in the pessary therapy (mean difference=−€1807, 95% CI −€2172; −€1446 and mean difference=−€1850, 95% CI −€2349; −€1341, respectively). The probability that pessary therapy is cost-effective compared with surgery was 1 at willingness-to-pay thresholds between €0 and €20 000/QALY gained from both perspectives.

**Conclusions** Non-inferiority of pessary therapy regarding the PGI-I could not be shown and no statistically significant differences in QALYs between interventions were found. Due to significantly lower costs, pessary therapy is likely to be cost-effective compared with surgery as an initial

## STRENGTHS AND LIMITATIONS OF THIS STUDY

⇒ This economic evaluation was performed alongside a multicentre pragmatic randomised controlled trial. The randomisation process ensures that groups are comparable and decrease the likelihood of selection bias, while the multicentre pragmatic design improves generalisability of results and transferability to clinical practice.

⇒ Validated outcome measures were used and the trial had a long-term follow-up of 2 years.

⇒ Consultations related to both interventions were provided by gynaecologists, which may overestimate intervention costs, as these consultations may be provided by trained general practitioners at lower costs.

⇒ Resource utilisation related to the specific medical treatment of interventions' complications (eg, medications), productivity costs related to unpaid work and informal care costs were not available and, thus, not included in the analysis, which may underestimate total costs.

⇒ Costs were estimated based on the Dutch reimbursement system and can differ from countries which may hamper the generalisability of results to healthcare systems in other countries.

treatment option for women with symptomatic POP treated in secondary care.

**Trial registration number** NTR4883.

## INTRODUCTION

Pelvic organ prolapse (POP) is a gynaecological condition in which one or more of the pelvic organs (ie, uterus, rectum, bladder, small bowel) herniate into the vagina due to weakness or damaging of the pelvic floor muscles and ligaments.[1 2] POP symptoms (eg, urinary, bowel and sexual dysfunction) are associated with decreased quality of life.[3] The

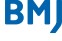

estimated prevalence of patient-reported POP symptoms ranges from 3% to 17.7% and is expected to increase with an ageing population. As a result, the demand for care and associated costs are also expected to increase.[4]

Effective treatment options for moderate to severe POP symptoms include pessary therapy and surgery.[5][6] However, both treatment options are not equally effective since non-inferiority of pessary therapy compared with surgery has not been shown.[7] A pessary is a silicone flexible device that is inserted into the vagina to support the pelvic organs (ie, uterus and bladder).[8] An advantage of pessary therapy is its minimally invasive nature. However, adverse effects (eg, discomfort, pain or excessive discharge) may occur in up to 49% of women within 12–24 months after fitting a pessary.[9][10] As for the surgery procedure, side-effects may include urinary tract infection and urinary bladder retention which may lead to longer hospital stay admission.[7] A recent observational study in women with a strong treatment preference and a randomised controlled trial (RCT) in women without a preference found a high crossover rate from pessary therapy to surgery of 24% and 54%, respectively.[7][9] Consequently, using pessary therapy as an initial treatment option might delay effective treatment, thereby increasing the demand for care and, thus, healthcare costs. However, using a pessary as a first treatment step would prevent expensive surgery if the pessary therapy relieves women symptoms adequately, making the initial use of pessary therapy potentially cost-effective compared with immediate surgery.

According to a recent systematic review,[8] only one model-based economic evaluation based on data from the USA conducted more than 10 years ago compared the cost-effectiveness of expectant management, pessary therapy, and surgery for POP symptoms.[11] This review reported that both pessary therapy and surgery were cost-effective compared with expectant management.[11] The aim of this study was to further investigate the cost-effectiveness of initial pessary therapy compared with immediate surgery from a healthcare and a societal perspective for moderate to severe POP symptoms with 2 years of follow-up. This study was performed alongside a non-inferiority randomised trial, of which the results have recently been published.[7]

## METHODS

### Study design

An economic evaluation was conducted alongside a non-inferiority RCT comparing pessary therapy and surgery as an initial treatment for moderate to severe POP in secondary care, the PEOPLE Project. The health economic analysis plan is available in the study protocol provided as online supplemental file 1. Participants were recruited between March 2015 and November 2019; the follow-up ended in June 2022. Detailed information about the PEOPLE Project is published elsewhere.[7][9][12] No substantial changes were made to the protocol after the commencement of the RCT.[7][12] This economic evaluation is reported according to the Consolidated Health Economic Evaluating Reporting Standards statement.[13]

### Study population

Women with POP symptoms who were referred by their general practitioner (GP) to secondary care were eligible for participation.[7] Inclusion criteria were POP stage ≥2 according to the Pelvic Organ Prolapse Quantification (POP-Q) system[14] and moderate to severe POP symptoms, defined as a prolapse domain score of >33 on the validated original Urinary Distress Inventory (UDI-6).[15] Exclusion criteria were prior prolapse or incontinence surgery, probability of future childbearing, insufficient knowledge of the Dutch language, comorbidity causing increased surgical risks, major psychiatric illness and prior pessary use.[7] Participants had to successfully complete a 30-minute pessary fitting trial to be eligible for randomisation. After informed consent was signed, participants were randomly allocated to either pessary therapy or surgery in a 1:1 ratio.[7] Randomisation used random permuted block sizes of 2 and 4 and was stratified by centre. Due to the nature of the treatment, treatment allocation was not concealed. Women who actively opted for a treatment were asked to participate in an observational cohort performed alongside the RCT; their data were not included in economic evaluation but published in another article.[9] Detailed information about study design and randomisation can be found elsewhere.[7][12]

### Setting and location

21 Dutch hospitals participated in this multicentre RCT. In the Netherlands, women with moderate to severe POP symptoms are generally referred to secondary care. Treatment options in secondary care include pessary therapy or surgery, which are both reimbursed by the Dutch healthcare system. All gynaecologists fitted at least 100 pessaries and performed 100 POP surgeries prior to study initiation.

### Comparators

#### Pessary therapy

Two main types of pessary therapy were offered to participants, namely, supportive (ie, ring) and occlusive (ie, space filling).[16] The pessary fitting was considered successful if the patient felt comfortable with the pessary in situ and if there was no pessary expulsion 30 min after fitting.[7] All women received verbal and written instructions on self-management of pessary therapy.[7] If self-management was not possible or preferred, an additional follow-up consultation with their gynaecologist or GP was scheduled every 4 months for pessary cleaning and vaginal inspection.[7] In case women performed self-management, the frequency of cleaning was left to their personal preference; however, it was advised to clean their pessary at least every 4 months. Women were instructed to return to the hospital if they experienced any symptoms or adverse events due to pessary therapy.[7]

## Surgery

Surgical intervention included a range of surgical procedures for the correction of three main types of prolapse that can occur individually or simultaneously, namely, (1) uterine descent, (2) cystocele and/or (3) rectocele.[7] For a cystocele or rectocele, respectively, a conventional anterior or posterior colporrhaphy was the standard technique. For a uterine descent, uterine-preserving techniques or a vaginal hysterectomy was performed.[7] All surgical interventions were performed following Dutch guidelines recommendations.[7 17] Decisions on which surgical technique was performed were decided in a shared decision manner between the gynaecologist and participant.[7] Women were instructed to return to the hospital if they experienced any symptoms or adverse events.

## Study perspective, time horizon and discount rate

This economic evaluation was conducted from a healthcare and a societal perspective over a time horizon of 24 months based on the literature and as recommended by the National Institute for Health and Clinical Excellence.[6 8 18] The healthcare perspective included costs related to interventions (pessary therapy and surgery) and healthcare utilisation costs. The societal perspective included costs related to absenteeism from paid work in addition to the interventions' costs and healthcare utilisation costs. Discount rates of 1.5% and 4% were applied to quality-adjusted life-year (QALY) and costs, respectively, after the first year of the RCT as recommended by the Dutch Guideline for Economic Evaluations in healthcare.[19]

## Outcomes

### Health outcomes

Two health outcomes were used for the trial-based economic evaluation: patient-reported subjective improvement and QALYs. Subjective improvement was measured with the Patient Global Impression of Improvement (PGI-I)[20] Scale at 12-month and 24-month follow-up. The PGI-I is a single-question, 7-point Likert response scale ranging from 'very much worse' to 'very much better'.[20] Subjective improvement was defined as a response of 'much better' or 'very much better'.[21] The PGI-I is a validated, easy-to-apply questionnaire, and it strongly correlates with other validated outcome measures such as the POP-Q system.[14 20] The primary analysis of PGI-I compared with surgery was presented in a previous publication in which its non-inferiority could not be shown.[7] This secondary analysis was performed as planned in the study protocol (online supplemental file 1).[22]

The QALY incorporates the impact of interventions on both the quantity and quality of life.[23] It is a routinely used health outcome measure in economic evaluations because it allows decision-makers to compare the cost-effectiveness of a range of interventions for different health conditions.[23] In this study, QALYs were calculated based on the EQ-5D-3L data collected at baseline,

3-month, 6-month, 12-month and 24-month follow-up. The EQ-5D-3L includes five dimensions of quality of life (ie, mobility, self-care, usual activities, pain/discomfort and anxiety/depression) with three response levels (ie, no problems, some problems or extreme problems/unable to) describing 243 health states.[24] The participants' health states obtained from EQ-5D-3L responses were converted into utility values using the Dutch tariff.[25] The utility values were used to calculate QALYs by means of linear interpolation (ie, the duration of a health state is multiplied by the utility related to that health state).[26]

### Cost outcomes

All costs were indexed to 2022 using the consumer price index in the Netherlands (www.cbs.nl).[27]

### Intervention costs

Intervention costs of the pessary therapy included those related to the pessary device and one gynaecologist consultation for the pessary placement at baseline. Unit prices of pessary therapy were based on the Dutch costing guideline[28] and on market prices (online supplemental file 2). For the surgery group, intervention costs consisted of the surgical procedures conducted at baseline. Unit prices of surgical procedures were based on the Diagnosis Treatment Combination (in Dutch, Diagnose Behandeling Combinatie (DBC)).[29] The DBC is a care path that includes diagnostic procedures and care activities delivered at hospital and immediate follow-up up to 6 weeks (42 days).[29] The average national prices are calculated for each DBC code based on all declared reimbursements that have been submitted to the DBC Information System by healthcare providers in hospital care. A detailed description of the resources used in the interventions and their respective unit costs is presented in online supplemental file 2.

### Healthcare utilisation costs

Healthcare utilisation was collected during follow-up visits at hospital centres including information on the number of scheduled consultations with gynaecologists and extra consultations due to complications, the number of days of hospital readmissions due to complications, the type/number of surgeries after pessary, the type/number of resurgeries, the number of times a pessary device was changed and the use of a pessary after initial surgery. Additionally, an adapted version of the institute for Medical Technology Assessment (iMTA) Medical Consumption Questionnaire[30] was used to measure non-intervention-related healthcare utilisation at 3-month, 6-month, 12-month and 24-month follow-up. Healthcare utilisation included resources used in primary care (ie, the number of GP consultations and other healthcare professionals due to POP symptoms) and in secondary care apart from study-scheduled consultations (ie, the number of extra consultations with other medical specialists due to POP symptoms). The number of healthcare resources used was then multiplied by their respective unit prices. Unit

**Table 1** Baseline characteristics of participants

| Baseline characteristics | Pessary therapy n=218 | Surgery n=221 |
|---|---|---|
| Age, mean (SD) | 64.8 (9.5), n=218 | 64.7 (9.2), n=221 |
| Risk-increasing aspects*, n, (%) | 71 (32.6), n=218 | 58 (26.2), n=221 |
| History of gynaecological surgery, n (%) | 22 (10.1), n=218 | 28 (12.7), n=221 |
| Family history of prolapse, n (%) | 106 (48.6), n=218 | 107 (49.5), n=216 |
| Parity, median (IQR) | 2.0 (2–3), n=215 | 2.0 (2–3), n=220 |
| Postmenopausal, n (%) | 186 (92.5), n=201 | 185 (90.2), n=205 |
| Duration of symptoms in months, median (IQR) | 6 (2–24), n=211 | 6 (3–24), n=216 |
| Vaginal atrophy, n (%) | 106 (56.7), n=187 | 110 (57.3), n=192 |
| Prolapse stage, n (%) | | |
| II (moderate) | 85 (39.0), n=218 | 102 (46.2), n=221 |
| ≥III (severe) | 133 (61.0), n=218 | 119 (53.9), n=221 |
| PGIS score, n (%) | | |
| I (not severe) | 13 (6.3), n=205 | 9 (4.4), n=205 |
| II (mild) | 48 (23.4), n=205 | 50 (24.4), n=205 |
| III (moderate) | 99 (48.3), n=205 | 112 (54.6), n=205 |
| IV (severe) | 45 (22.0), n=205 | 34 (16.6), n=205 |
| PFDI-20 score†, n (%) | | |
| POPDI-6 score | 29.5 (19.2), n=210 | 28.7 (15.6), n=208 |
| CRADI-8 score | 13.9 (15.1), n=210 | 12.1 (12.6), n=208 |
| UDI-6 score | 26.0 (22.0), n=209 | 25.2 (20.0), n=208 |
| PFDI-20 total score | 69.3 (45.7), n=209 | 65.9 (37.7), n=208 |
| EQ-5D utility value‡, mean (SD) | 0.87 (0.15), n=209 | 0.85 (0.15), n=206 |

*Presence of one or more comorbidities: smoking, use of antidepressants, obesity, diabetes mellitus, chronic pulmonary disease.
†PFDI-20: the subscale scores range from 0 to 100 and the total score ranges from 0 to 300. Higher scores indicate more symptom distress.
‡EQ-5D utility values: the Dutch EQ-5D tariffs range from −0.33 to 1.
%, proportion; CRADI-8, Colorectal-Anal Distress Inventory; IQR, interquartile range; n, number of women; PFDI-20, Pelvic Floor Distress Inventory; PGIS, Patient Global Impression of Severity; POPDI-6, Pelvic Organ Prolapse Distress Inventory; SD, standard deviation; UDI-6, Urinary Distress Inventory.

prices of healthcare resources were based on the Dutch costing guideline[28] (online supplemental file 2).

### Lost productivity costs

Absenteeism from paid work due to POP symptoms was measured using an adapted version of the iMTA Productivity Cost Questionnaire[31] at 3-month, 6-month, 12-month and 24-month follow-up. The friction cost approach (FCA) was used to calculate sickness absenteeism costs related to paid work.[32] The FCA assumes that sickness absenteeism costs are limited to the period needed to replace an absent sick worker (the friction period), which has been estimated to be 12 weeks (85 days) in the Netherlands.[32] Gender-specific estimates of the mean wages of the Dutch population were used to calculate sickness absenteeism costs from paid work.[28]

### Cost-effectiveness analysis

Analyses were performed according to the intention-to-treat principle using StataSE V.17. As recommended by Faria et al,[33] mean imputation was used to impute missing

values at baseline (ie, parity, Patient Global Impression of Severity, Pelvic Floor Distress Inventory (PFDI-20), Pelvic Organ Prolapse Distress Inventory (POPDI-6), Colorectal-Anal Distress Inventory (CRADI-8), UDI-6 and EQ-5D utility values). Subsequently, multiple imputation by chained equations was used to impute follow-up missing data. The multiple imputation model included treatment group and hospital centre, variables associated with missingness (ie, body mass index, number of resurgeries, number of consultations and family history of prolapse), outcomes and potential confounders (ie, age, history of gynaecological operations, prolapse stage, menopausal state and risk-increasing aspects).[34] Risk-increasing aspects were a combined variable that included at least one of the following comorbidities: smoking status, antidepressants use, obesity, diabetes mellitus and chronic pulmonary disease. Predictive mean matching was used in the imputation procedure to account for the skewed distribution of the costs.[35] Missing cost data were imputed at the level of resource use by time point (ie, number of

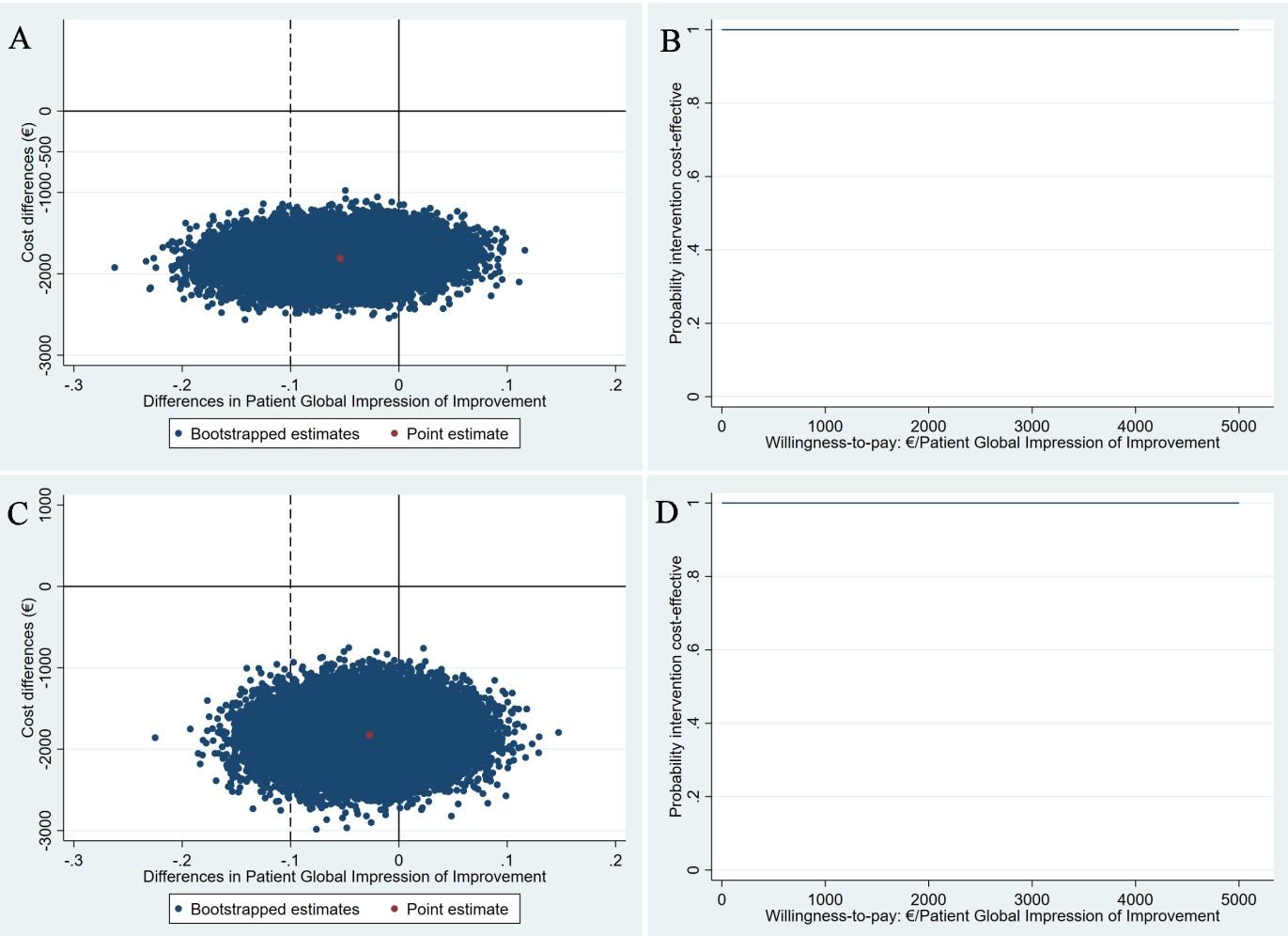

**Figure 1** Cost-effectiveness planes (CE-planes) and cost-effectiveness acceptability curves (CEACs) for Patient Global Impression of Improvement (PGI-I). CE-planes (A,B) and CEACs (C,D) comparing pessary therapy with surgery for the PGI-I outcome from a healthcare and a societal perspective, respectively. CE-planes show the incremental cost-effectiveness ratio point estimate (red dot) and the distribution of the 5000 replications of the bootstrapped cost–effect pairs (blue dots). CEACs indicate the probability of pessary therapy being cost-effective compared with surgery (y-axis) for different willingness-to-pay (WTP) thresholds per unit of PGI-I gained (x-axis). The dashed line represents the non-inferiority margin of 10%. (A and C) All of bootstrapped cost–effect pairs were distributed in the southern quadrants of the CE-planes, meaning that the pessary therapy is less costly but could also be less and more effective. 83.2% bootstrapped cost–effect pairs are situated on the right of the non-inferiority margin for effects. (B and D) A steady probability of 1 that the pessary therapy is cost-effective compared with surgery for different WTP thresholds per PGI-I gained. PGI-I is presented as the difference between groups in the proportion of participants reporting improvement.

consultations, working hours and absenteeism hours). The number of imputations was increased until there was a loss of efficiency of ≤5%, resulting in 10 imputed datasets.[36] The 10 imputed datasets were analysed separately and estimates were pooled using Rubin's rules.[37]

Multilevel linear regression models were used to estimate the difference in costs and effects between the groups to account for the fact that randomisation was stratified by hospital centre.[38] For cost and effect outcomes, a two-level structure was used where participants and hospital centre represented the first and second level, respectively. All analysis models were adjusted for relevant baseline confounders. The PGI-I model was adjusted for risk-increasing aspects and prolapse stage. The QALY model was adjusted for baseline utility values,[39] risk-increasing

aspects and prolapse stage. Healthcare and societal costs models were adjusted for age, menopause state, risk-increasing aspects and prolapse stage. A non-inferiority margin of 10% risk difference (one-sided 95% CI) was set for the PGI-I outcome based on the expectation that 80% of women would report successful treatment (either pessary therapy or surgery) after 2 years.[12 40 41]

Incremental cost-effectiveness ratios (ICERs) were calculated by dividing the difference in costs between the pessary therapy and surgery by their difference in effects resulting in an estimate of the costs per unit of effect gained. Bias-corrected accelerated bootstrapping with 5000 replications was used to estimate the joint uncertainty surrounding differences in costs and effects. Bootstrapped cost–effect pairs were described and plotted on

Table 2  Effects and costs by treatment group and difference at 24-month follow-up

| | Pessary therapy n=218 | Surgery n=221 | Unadjusted difference (95% CI) |
|---|---|---|---|
| **Effects** | | | |
| PGI-I, n (%) | 164 (75.1) | 179 (80.8) | −0.06 (−0.15; 0.04) |
| QALY, mean (SE) | 1.80 (0.02) | 1.82 (0.01) | −0.02 (−0.06; 0.02) |
| **Costs, mean (SE)** | | | |
| Intervention costs | 178 (0.2) | 4640 (0) | −4462 (−4463; −4462) |
| Primary care costs | 18 (2) | 15 (2) | 3 (−3; 8) |
| Secondary care costs | 3736 (174) | 1127 (80) | 2609 (2232; 2982) |
| Healthcare costs | 3932 (174) | 5782 (80) | −1850 (−2228; −1476) |
| Absenteeism from paid work | 362 (117) | 390 (120) | −28 (−338; 290) |
| Societal costs | 4294 (227) | 6172 (150) | −1878 (−2395 to to 1345) |

Intervention costs in the pessary group=costs of pessary device and pessary placement consultation at baseline. Intervention costs in the surgery group=DBC costs of surgery at baseline which included one follow-up consultation at 6 weeks. Primary care costs=costs of general practitioner or other healthcare professional consultations apart from the prescheduled follow-up consultations because of complaints related to pelvic organ prolapse symptoms. Secondary care costs=costs of follow-up scheduled consultations with gynaecologists attended by patients and extra consultations due to complications, costs of hospital readmissions due to complications, surgeries after pessary, resurgeries and costs of pessary change.
PGI-I is presented as the difference between groups in the proportion of participants reporting improvement.
%, proportion; DBC, Diagnose Behandeling Combinatie; n, number of participants; PGI-I, Patient Global Impression of Improvement (1=improvement; 0=no improvement); QALY, quality-adjusted life-year; SE, standard error.

cost-effectiveness planes (CE-planes).[42] Non-inferiority with regard to cost-effectiveness was demonstrated using a one-sided α of 2.5%, meaning that 97.5% of the cost–effect pairs have to lie right of the non-inferiority margin for effects.[43] Cost-effectiveness acceptability curves (CEACs) were estimated to show the probability of the pessary therapy being cost-effective compared with surgery for a range of willingness-to-pay (WTP) thresholds (ie, the maximum amount of money society is willing to pay for a unit of effect).[44] For QALY, we used a WTP threshold of €20 000/QALY gained recommended by the Dutch Health Care Institute.[45] As there is no specific WTP threshold for PGI-I, we used a maximum WTP of €5237/PGI-I gained. This threshold was based on the average DBC costs of surgical procedures performed for POP symptoms as reported in online supplemental file 2.

### Sensitivity analysis

Two sensitivity analyses (SAs) were performed to assess the robustness of the results. SA1 was a complete case analysis, meaning that only observations with complete data were included in the main analysis. A per-protocol analysis (SA2) was performed to compare treatment groups including women who completed the treatment with which they were originally allocated.

### Patient and public involvement

One major gynaecological patient organisation in the Netherlands (ie, BekkenBodem4All) as well as the Dutch Urogynecology Consortium fully agreed on the study protocol and identified the study as highly relevant.[12]

## RESULTS

### Participants

Of the 1605 women assessed for eligibility, 440 were randomised to either pessary therapy (n=218) or surgery (n=222) as shown in online supplemental file 2. After randomisation, one participant was excluded from the surgery group due to prolapse stage 1 resulting in a total of 221 women in this group (online supplemental file 2). Baseline incomplete data were imputed for parity (n=4, 0.9%), PFDI-20 (n=22, 5.0%), POPDI-6 (n=21, 4.8%), CRADI-8 (n=21, 4.8%), UDI-6 (n=22, 5.0%) and utility values (n=24, 5.5%) (table 1). Follow-up missing data at 24 months were multiply imputed for PGI-I (n=104, 23.7%), QALY (n=144, 32.8%), healthcare costs (n=160, 36.4%) and societal costs (n=165, 37.6%) (figure 1). A total of 118 of 218 (54.1%) women randomised to pessary therapy crossed over to surgery, and a total of 8 women out of 221 (3.6%) underwent recurrent surgery. At baseline, no meaningful differences were found between both groups (table 1).

### Effectiveness

In the unadjusted analysis, the lower 95% CI bound of the PGI-I outcome surpassed the non-inferiority margin of −10% (mean difference −0.06, 95% CI −0.15; 0.04), meaning that non-inferiority of pessary therapy compared with surgery could not be shown (table 2). After adjusting for confounders, the lower 95% CI bound of the PGI-I outcome still surpassed the non-inferiority margin (mean difference −0.05, 95% CI −0.14; 0.03, table 3). There was no statistically significant difference in QALYs between groups neither in the unadjusted analysis (mean

**Table 3** Results of the cost-effectiveness (CE) and cost-utility analysis

| Effect outcome | ΔE (95% CI) | ΔC (95% CI) | ICER | Proportion of bootstrapped cost–effect pairs in the CE-plane | | | |
|---|---|---|---|---|---|---|---|
| | | | | NE | SE | SW | NW |
| Main analysis—healthcare perspective | | | | | | | |
| PGI-I, n=439 | −0.05 (−0.14; 0.03) | −1807 (−2172; −1446) | 33 509 | 0% | 9% | 91% | 0% |
| QALY, n=439 | −0.03 (−0.07; 0.002) | −1807 (−2172; −1446) | 52 980 | 0% | 3% | 97% | 0% |
| Main analysis—societal perspective | | | | | | | |
| PGI-I, n=439 | −0.05 (−0.14; 0.03) | −1850 (−2349; −1341) | 34 295 | 0% | 9% | 91% | 0% |
| QALY, n=439 | −0.03 (−0.07; 0.002) | −1850 (−2349; −1341) | 54 223 | 0% | 3% | 97% | 0% |
| Sensitivity analysis 1: complete case analysis—healthcare perspective | | | | | | | |
| PGI-I, n=259 | −0.02 (−0.11; 0.07) | −1976 (−2460; −1585) | 81 560 | 0% | 25% | 75% | 0% |
| QALY, n=256 | −0.01 (−0.05; 0.03) | −1962 (−2470; −1572) | 236 907 | 0% | 33% | 67% | 0% |
| Sensitivity analysis 1: complete case analysis—societal perspective | | | | | | | |
| PGI-I, n=254 | −0.02 (−0.11; 0.08) | −1884 (−2499; −1241) | 99 339 | 0% | 30% | 70% | 0% |
| QALY, n=252 | −0.005 (−0.05; 0.04) | −1860 (−2500; −1225) | 367 444 | 0% | 39% | 61% | 0% |
| Sensitivity analysis 2: per-protocol analysis—healthcare perspective | | | | | | | |
| PGI-I, n=271 | −0.13 (−0.25; −0.01) | −4398 (−4583; −4311) | 33 044 | 0% | 1% | 99% | 0% |
| QALY, n=271 | −0.01 (−0.05; 0.02) | −4398 (−4583; −4311) | 358 020 | 0% | 27% | 73% | 0% |
| Sensitivity analysis 2: per-protocol analysis—societal perspective | | | | | | | |
| PGI-I, n=271 | −0.13 (−0.25; −0.01) | −4748 (−5159; −4498) | 35 676 | 0% | 1% | 99% | 0% |
| QALY, n=271 | −0.01 (−0.05; 0.02) | −4748 (−5159; −4498) | 386 539 | 0% | 27% | 73% | 0% |

ΔC=difference in costs in €; ΔE=difference in effects; ICER=€ per unit of effect gained; CE-plane=CE plane showing the difference in costs between pessary therapy and surgery on the y-axis and the difference in effects on the x-axis resulting in four quadrants, namely, NE=pessary therapy more expensive and more effective than surgery; SE=pessary therapy less expensive and more effective than surgery; SW=pessary therapy less expensive and less effective than surgery; NW=pessary therapy more expensive and less effective than surgery. The PGI-I model was adjusted by risk-increasing aspects and prolapse stage. The QALY model was adjusted by baseline utility values, risk-increasing aspects and prolapse stage. Healthcare and societal costs models were adjusted by age, menopause state, risk-increasing aspects and prolapse stage. PGI-I is presented as the difference between groups in the proportion of participants reporting improvement.
ICER, incremental cost-effectiveness ratio; NE, northeast; NW, northwest; PGI-I, Patient Global Impression of Improvement; QALY, quality-adjusted life-year; SE, southeast; SW, southwest.

difference −0.02, 95% CI −0.06; 0.02, table 2) nor the adjusted analysis (mean difference −0.03, 95% CI −0.07; 0.002, table 3).

## Costs

After 24 months, unadjusted analyses showed there were statistically significant savings in the pessary therapy group compared with the surgery for both total healthcare costs (mean difference −€1850, 95% CI −€2228; −€1476) and societal costs (mean difference −€1878, 95% CI −€2395; −€1345) (table 2). Despite having other surgery options (online supplemental file 2), we used a fixed price of €4640 considering the surgical procedures conducted in the trial. The main cost driver in the surgery group was the intervention costs (€4640, SE=0), while in the pessary therapy group, this was secondary costs (€3736, SE=174) (table 2). Given that half of patients in the pessary group crossed over to surgery (54.1%) and a small proportion of women underwent recurrent surgery in the surgery group (3.6%), secondary costs during follow-up were statistically significantly higher in the pessary therapy group compared with surgery (mean difference €2609,

95% CI €2232; €2982, table 2). In the adjusted analysis, mean differences in healthcare and societal costs between groups slightly decreased compared with the unadjusted analysis (table 3). However, both healthcare and societal costs in the pessary group were still statistically significantly lower than in the surgery group.

## Cost-effectiveness analysis

For the PGI-I outcome, the main analysis showed ICERs of 33 509 and 34 295 from a healthcare and a societal perspective, respectively (table 3). The positive ICERs are situated in the southwest quadrant of the CE-plane and indicate that while pessary therapy incurred significantly lower costs (healthcare mean difference −€1807, 95% CI −€2172; −€1446 and societal mean difference −€1850, 95% CI −€2349; −€1341), it was also less effective compared with surgery (mean difference=−0.05, 95% CI −0.14; 0.03), although not statistically significantly so. Most bootstrapped cost–effect pairs were situated on the right of the non-inferiority margin for effects (83.2%) and in the southern quadrants of the CE-plane, meaning

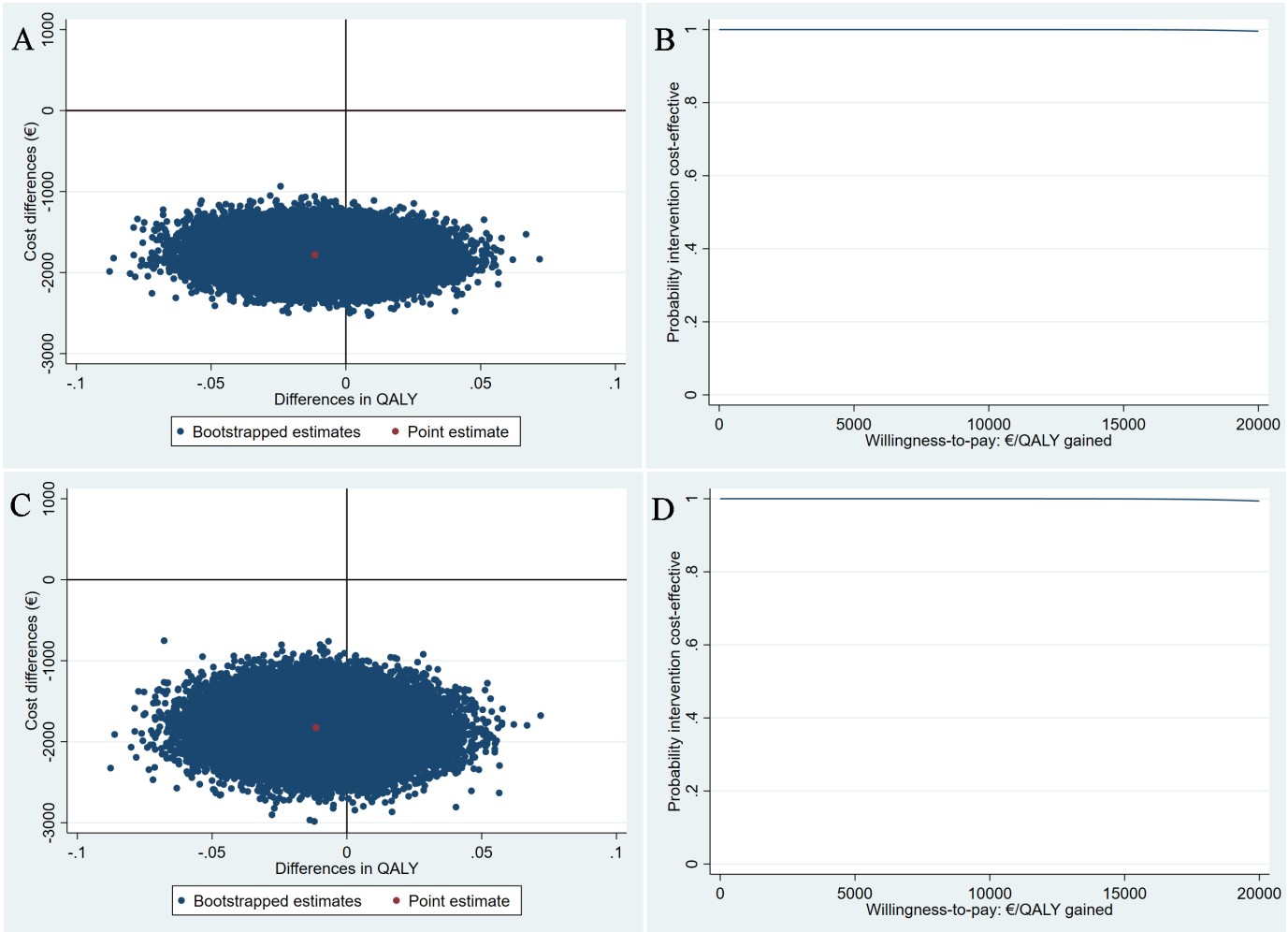

**Figure 2** Cost-effectiveness planes (CE-planes) and cost-effectiveness acceptability curves (CEACs) for quality-adjusted life-years (QALYs). CE-planes (A and C) and CEACs (B and D) comparing pessary therapy with surgery for QALY from a healthcare and a societal perspective, respectively. CE-planes show the incremental cost-effectiveness ratio point estimate (red dot) and the distribution of the 5000 replications of the bootstrapped cost–effect pairs (blue dots). CEACs indicate the probability of pessary therapy being cost-effective compared with surgery (y-axis) for different willingness-to-pay (WTP) thresholds per QALY gained (x-axis). (A and C) All of bootstrapped cost–effect pairs were distributed in the southern quadrants of the CE-planes, meaning that the pessary therapy is less costly but could also be less and more effective. (B and D) A steady probability of 1 that the pessary therapy is cost-effective compared with surgery for different WTP thresholds per QALY gained.

that pessary therapy would save costs at an acceptable loss of effect in terms of PGI-I (figure 1A and C). Due to statistically significant lower healthcare and societal costs in the pessary therapy group compared with surgery, CEACs showed that the probability of the pessary therapy being cost-effective compared with surgery was 1 at relevant WTP values (figure 1B and D). This means that the pessary therapy as an initial treatment option has a 100% probability of being cost-effective compared with immediate surgery.

For QALYs, similar to PGI-I, the positive ICERs indicate that pessary therapy is less expensive and less effective (mean difference −0.03, 95% CI −0.07; 0.002) than surgery. However, the difference in QALYs was small and less than the commonly used minimally clinically important difference (ie, 0.06),[46 47] meaning that pessary therapy would save costs without considerably reducing health-related quality of life. The majority of the

bootstrapped cost–effect pairs were in the southern quadrants of the CE-plane (100%), meaning that the pessary therapy was less costly than surgery (figure 2A and C). The probability that pessary therapy being cost-effective compared with surgery at all WTP thresholds was 1 from both perspectives (figure 2B and D).

**Sensitivity analysis**

SA1 including only complete cases showed similar results compared with the main analysis (table 3). In SA2, which included women who received their originally allocated intervention with fully imputed data on the PGI-I (pessary therapy n=81, surgery n=190), the differences in costs and PGI-I between pessary and surgery increased and in QALY decreased compared with the main analysis (table 3). However, this did not affect the cost-effectiveness results.

## DISCUSSION
### Main findings
This economic evaluation showed that non-inferiority of pessary therapy compared with surgery with regard to subjective improvement could not be shown, which was consistent with primary analysis of PGI-I.[7] Also, there were no statistically significant differences in QALY gained. Despite this, a strategy of initial pessary therapy in women with symptomatic POP is likely to be cost-effective compared with immediate surgery from a healthcare and a societal perspective due to lower costs associated with pessary therapy.

### Explanation of the findings and comparison with the literature
For both effect outcomes, the high probability of pessary therapy being cost-effective compared with surgery is explained by the fact that total healthcare and societal costs in the pessary group were statistically significantly lower than in the surgery group, despite the high proportion of crossover (54.1%) from participants in the pessary group to surgery.

Recently, Bugge et al[8] systematically reviewed the (cost-)effectiveness of pessary therapy for managing POP symptoms and found only two economic evaluations.[11 48] Of those, only Hullfish et al[11] directly compared pessary therapy with surgery. They developed a model-based economic evaluation with 12-month follow-up based on data from the literature, local experience of a single institution and expert opinion. Results showed that for lower WTP thresholds (ie, from \$0 to \$5600/QALY gained), pessary is cost-effective compared with surgery and for higher WTP thresholds (ie, from \$5600 to roughly \$20 000/QALY gained) not anymore. Our results, based on randomised data, showed that pessary therapy is cost-effective compared with surgery at similar WTP thresholds (ie, €0–20 000/QALY gained).

### Strengths and limitations
One of the strengths of this study is that it was performed alongside a multicentre pragmatic RCT. The randomisation process ensures that groups are comparable and decrease the likelihood of selection bias,[49] while the multicentre pragmatic design improves generalisability of results and transferability to clinical practice. Validated outcome measures were used and the trial had a follow-up of 2 years. However, since POP symptoms can relapse over time, studies including a longer follow-up (eg, more than 2 years) are needed. This study has a number of limitations. First, productivity costs related to unpaid work such as number of hours spent in unpaid activities (eg, voluntary and housework) and informal care (eg, care provided by family and friends while being sick) were not collected. Since the mean age of the participants is 65 years (the retirement age in the Netherlands until 2024), these costs are likely to be more relevant than lost productivity related to paid work. Second, consultations related to both interventions were provided by gynaecologists, which may result in an overestimation of intervention costs. This may not be representative for healthcare systems in other countries, as these consultations may be provided by trained GPs at lower costs (ie, €39 by a GP vs €109 by a medical specialist). Third, healthcare resource utilisation related to the specific medical treatment of complications (eg, medications) was not collected. Only costs related to readmissions and extra complications due to complications were included in the analysis. This may underestimate healthcare utilisation costs. Fourth, the proportion of missing data on the outcomes was between 24% and 38%. To deal with this issue, multiple imputation of missing values was performed which is the recommended method to handle missing data in trial-based economic evaluations to produce valid estimates.[33 50 51] In addition, an SA including complete cases was performed to evaluate the robustness of findings, showing that results were not affected. Fifth, costs were estimated based on the Dutch reimbursement system and can differ from countries which may hamper the generalisability of results to healthcare systems in other countries.

### Implications for practice and future research
A considerable number of women declined to participate in the RCT (n=553, figure 1). These women were offered the possibility to participate in a prospective cohort.[9] The majority of participants in the prospective cohort opted for a pessary therapy as initial treatment option (62.2%).[9] Compared with participants of the RCT,[7] participants in the cohort less often crossed over to surgery (24% vs 54%). In addition, in this cohort, more women reported successful improvement after surgery compared with pessary.[9] This suggests that it is important to consider women's preferences when deciding about the most suitable treatment for their POP symptoms. Future studies should measure costs from a broader perspective than this study did, as relevant costs were not considered in the analysis, that is, costs related to follow-up medical treatment, informal care costs and lost productivity costs related to unpaid work (eg, housework, voluntary work).

## CONCLUSION
Non-inferiority of pessary therapy with regard to the PGI-I could not be shown and there were no statistically significant differences in QALYs between interventions. Due to significantly lower costs, pessary therapy is likely to be cost-effective compared with immediate surgery from a healthcare and a societal perspective as an initial treatment option for women with moderate to severe POP symptoms treated in secondary care. However, considering the high crossover rate from pessary therapy to surgery, it is important to consider women's preferences regarding the treatment of their POP systems.

**Author affiliations**
[1]Department of Health Sciences, Vrije Universiteit Amsterdam, Amsterdam, The Netherlands
[2]Department of Obstetrics and Gynaecology, Amsterdam Reproduction & Development Research Institute, University of Amsterdam, Amsterdam, The Netherlands

³Department of Health Sciences, VU University Amsterdam, Amsterdam, The Netherlands

⁴Department of Obstetrics and Gynecology, University of Amsterdam, Amsterdam, The Netherlands

⁵Department of Gynecology, Women's Health Bergman Clinics, Amsterdam, The Netherlands

⁶Department of Primary and Community Care, Radboud University Medical Center, Nijmegen, The Netherlands

⁷Department of Obstetrics and Gynecology, University Medical Center Utrecht, Utrecht, The Netherlands

⁸Department of Gynecology, Women's Health Bergman Clinics, Hilversum, The Netherlands

⁹Department of Obstetrics and Gynecology, Spaarne Hospital, Haarlem, The Netherlands

**Collaborators** PEOPLE group: Milani F, van Bavel J, Bon GG, Bongers MY, Bos K, Broekman AMW, Dietz V, van Eijndhoven HWF, Hakvoort RA, Janszen EWM, Kluivers K, Link G, Massop-Helmink DS, van der Ploeg JM, Sikkema FM, Spaans WA, van der Ster A, Verheijen-van de Waarsenburg K, Vernooij MMA, Weemhoff M, Weis-Potters AE.

**Contributors** ÂJB—methodology, data curation, formal analysis and writing (original draft). LRvdV—methodology, data curation, writing (review and editing) and project administration. JB—methodology, writing (review and editing) and supervision. J-PWRR—writing (review and editing). ALML-J—writing (review and editing). CHvdV—conceptualisation, methodology, resources, funding acquisition and writing (review and editing). AV—guarantor, conceptualisation, methodology, writing (review and editing) and supervision. All authors have agreed to the submission of the final paper.

**Funding** Financial support was provided through a personal grant (receiver: CHvdV) issued by the ZonMW, a Dutch governmental healthcare organisation. This study was funded on 26 June 2014 (project no 837002525).

**Competing interests** CHvdV reports grants from ZonMW, a Dutch government institution, during the conduct of the study.

**Patient and public involvement** Patients and/or the public were involved in the design, or conduct, or reporting, or dissemination plans of this research. Refer to the Methods section for further details.

**Patient consent for publication** Consent obtained directly from patient(s).

**Ethics approval** This study involves human participants. Ethical approval was obtained from the Medical Ethical Committee of University Medical Center Utrecht (METC 14-533/M) and registered at the Netherlands Trial Register (NTR) under number NTR4883. All participants provided written informed consent.

**Provenance and peer review** Not commissioned; externally peer reviewed.

**Data availability statement** Data are available upon reasonable request.

**ORCID iD**
Ângela J Ben http://orcid.org/0000-0003-4793-9026

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
