## [Reviewer comments · BMJ Open]

ARTICLE DETAILS

TITLE (PROVISIONAL)	Cost-effectiveness of pessary therapy versus surgery for symptomatic pelvic organ prolapse: an economic evaluation alongside a randomized non-inferiority controlled trial
AUTHORS	Ben, Ángela; van der Vaart, Lisa R.; Bosmans, J; Roovers, Jan-Paul W.R.; Lagro-Janssen, Antoinette; van der Vaart, Carl H.; Vollebregt, Astrid

VERSION 1 – REVIEW

REVIEWER	Thomas, Sonia RTI International
REVIEW RETURNED	04-Aug-2023

GENERAL COMMENTS	This is a well written and thorough analysis of cost effectiveness of pessary versus surgery as first-line treatment for pelvic floor prolapse with a follow-up of 24 months. It is a secondary outcome for the randomized non-inferiority PEOPLE clinical trial. Relatively minor revisions are recommended: Analysis of PGI-I: Effectiveness is evaluated by the PGI-I and a QALY. However, non-inferiority of the PGI-I was the primary outcome of this trial and has been previously reported (ref 7). This paper completed a new non-inferiority analysis of PGI-I using a different sample size and analysis methods (using MICE to impute missing data) than the primary paper, and the results are prominently presented in the abstract, results, and conclusions. The text should clearly indicate that the primary analysis of PGI-I for the trial was in the first publication (and provide a summary of these results), and this analysis is secondary (and explain reasoning why it was done), and state that results are consistent with primary findings. Modeling methods: The models used to compare groups for effectiveness and cost outcomes included adjustment for some post-base outcomes, such as the 12-month PGI-I or extra consultations due to complications. It is odd and not standard to adjust for post-baseline outcomes, and the reasoning should be clearly justified by the author or removed. Please provide modeling type for the PGI-I difference in proportion improvement — for example, was this a logistic model? Minor: • Cost Methods text: Add to text that intervention and health care costs are 2022.• Sensitivity analysis methods: Given that this was a pragmatic design, and it was expected that participants in pessary would cross-over and received surgery, then a subset analysis based on
---

	participants that did not cross over is odd to refer to as a “Per-protocol” analysis, as opposed to a “subset analysis” of participants who did not seek further treatment due to treatment failure/dissatisfaction.  • Methods text for cost-effect analysis and Table 3: define how the ICER is calculated and provide units. • Tables 2 and 3, Figures 1 and 2: show more clearly in the table body and figure axes that the PGI-I is presented as difference in proportion that reported much better or very much better. • Table 2: please explain why the standard deviation for the surgery intervention is zero, given that different types of POP surgery have different costs. • Figures 1 and 2: the CE graphs would be helpful if the vertical cost axis contained zero, and lines show the zero axes, to clearly demark the quadrants of the cost-effectiveness plane. • Strengths and limitations section: text that consultations being done by gynecologists when usual care is for a primary care doctor: might be helpful to note that this is not necessarily true in health care systems of other countries.
--	--

REVIEWER	Buskens, Erik University of Groningen, University Medical Center Groningen, Epidemiology
REVIEW RETURNED	11-Oct-2023

GENERAL COMMENTS	A very minor point regarding this otherwise very clear report is that the likelihood of initial pessary treatment being cost-effective is 1 whereas the authors conclude 'pessary therapy is likely to be cost-effective compared to surgery', which is not in line. Why not boldly state 'initial pessary treatment is the preferred treatment based on the balance between costs and effects' (or something along these lines)? The fact that many do subsequently cross over is not a good reason to weaken this conclusion. This might be the 'step up' that would still save costs without losing on QoL etc.
---

REVIEWER	Kilonzo, M University of Aberdeen
REVIEW RETURNED	18-Oct-2023

GENERAL COMMENTS	The paper is well written and follows the CHEERS checklist. There is one issue that could be presented in a clearer way. The results of the CEAC are mainly in the SW quadrant. Although the benefit difference is not significant it is on average less for the pessary group. The authors should explain what this means in terms of the fact that it is not the willingness to pay for an additional QALY but more the willingness to accept the loss in benefit. On page 24 Table 3 reports the different percentages of bootstrapped results in each of the different quadrants without explaining what the cost effectiveness plane is. The paper could benefit from some discussion and interpretation of the ICERs as some are negative and the values are much higher than the willingness to pay threshold. On page 15 line 43/44 the authors report that pessary is cost effective at all WTP thresholds. Do they mean the thresholds they refer to? The figure values do not exceed the WTP threshold applied it would be worthwhile for the reader if higher values were included in the figures. The conclusions of the study should be placed in the context that the study is a short term study (24 months) and there is need for longer
--

	term follow-up as the prolapse symptoms can reappear and there is need for longer term follow-up studies. Minor comments In the results section page 13 line 11 and line 21 there is reference to figure 1 did the authors mean supplementary figure 1 the consort diagram rather than the CEAC? Authors should check Page table 1 line 9/10 the n values are the wrong way round. The CEACs are very hard to decipher. The point estimate is barely visible . The authors should consider including the WTP lines for more clarification as the vertical axes do not include 0 and positive values Also, they should consider incorporating all four quadrants of the cost effectiveness plane as they report them in the text. Checklist comments In the checklist the authors state that an HEAP was developed and reference page 5 but there is no mention of it in the study. There authors have not included a justification for the study horizon. Although the authors include the cost year for the productivity costs there is no statement about all the other costs that are included in the study. The study does not include an economic model so questions 16,17.18 , 19 and 22 are not relevant to this study. No subgroup analysis was conducted and the authors should state this and justify why as there were some planned for the clinical effectiveness.
--	--

VERSION 1 – AUTHOR RESPONSE

Response to the Reviewers' comments

Comments of Reviewer #1	Responses
This is a well written and thorough analysis of cost effectiveness of pessary versus surgery as first-line treatment for pelvic floor prolapse with a follow-up of 24 months. It is a secondary outcome for the randomized non-inferiority PEOPLE clinical trial. Relatively minor revisions are recommended: Analysis of PGI-I: Effectiveness is evaluated by the PGI-I and a QALY. However, non-inferiority of the PGI-I was the primary outcome of this trial and has been previously reported (ref 7). This paper completed a new non-inferiority analysis of PGI-I using a different sample size and analysis methods (using MICE to impute missing data) than the primary paper, and the results are prominently presented in the abstract, results, and conclusions. The text should clearly indicate that the primary analysis of PGI-I for the trial was in the first publication (and provide a summary of these results), and this analysis is secondary (and explain	Thank you for pointing this out. This information has been added to the Methods and Discussion sections as follows: Page 7: “The primary analysis of PGI-I compared with surgery was presented in a previous publication in which its non-inferiority could not be shown⁷. This secondary analysis was performed as planned in the study protocol (Supplementary file 1)²¹.” Page 16: “Main findings This economic evaluation showed that although non-inferiority of pessary therapy with regard to subjective improvement could not be shown which was consistent with primary analysis of PGI-I⁷. Also, there were no statistically significant differences in QALY gained. Despite this, a strategy of initial pessary therapy in women with symptomatic POP is likely to

reasoning why it was done), and state that results are consistent with primary findings.	be cost-effective compared to immediate surgery from a healthcare and a societal perspective due to lower costs associated with pessary therapy.
Modeling methods: The models used to compare groups for effectiveness and cost outcomes included adjustment for some post-base outcomes, such as the 12-month PGI-I or extra consultations due to complications. It is odd and not standard to adjust for post-baseline outcomes, and the reasoning should be clearly justified by the author or removed.	As suggested, this information has been added to the Methods Section to clarify that covariates, the 12-month PGI-I and extra consultations due to complications, were included in the models as confounders as they were related to the outcomes and exposure. Page 10: “PGI-I at 12-month and extra consultations due to complications, were included in the models as confounders as they were related to the outcomes and treatment allocation.”
Please provide modeling type for the PGI-I difference in proportion improvement — for example, was this a logistic model?	The modelling technique used was a multilevel regression model as stated in the Methods section. Page 10: “Multilevel regression models were used to estimate the difference in costs and effects between the groups to account for the fact that randomization was stratified by hospital center³⁷.”
• Cost Methods text: Add to text that intervention and health care costs are 2022.	This information was included at the end of the paragraph describing the lost productivity costs, but it has now been moved up to the beginning of the paragraph describing the cost measures: Page 8: “All costs were indexed to 2022 using the consumer price index in the Netherlands(www.cbs.nl)³¹.”
• Sensitivity analysis methods: Given that this was a pragmatic design, and it was expected that participants in pessary would cross-over and received surgery, then a subset analysis based on participants that did not cross over is odd to refer to as a “Per-protocol” analysis, as opposed to a “subset analysis” of participants who did not seek further treatment due to treatment failure/dissatisfaction.	The sentence in the Methods section has been re-written to avoid confusion, although “Per-protocol” has been kept in the text for consistency to the terminology (Reference: Mo Y, Lim C, Watson JA, et al. Non-adherence in non-inferiority trials: pitfalls and recommendations. BMJ. 2020;370:m2215.). Pages 11: “Because we expected some participants to crossover from pessary to surgery, A per protocol analysis (SA2) was performed to compare treatment groups including women who completed the treatment to which they were originally allocated.”

 • Methods text for cost-effect analysis and Table 3: define how the ICER is calculated and provide units. 	As suggested, the ICER definition and units have been added to the text as follows: Page 10: “Incremental cost-effectiveness ratios (ICERs) were calculated by dividing the difference in costs between the pessary therapy and surgery by their difference in effects resulting in an estimate of the costs per unit of effect gained.” Bottom of Table 3: “ΔC= difference in costs in Euros; 95% CI = 95% confidence interval; ΔE= difference in effects; ICER = Incremental Cost-Effectiveness Ratio (€ per unit of effect gained)”
 • Tables 2 and 3, Figures 1 and 2: show more clearly in the table body and figure axes that the PGI-I is presented as difference in proportion that reported much better or very much better. 	As suggested, this information has been added at the bottom of Tables 2 and 3 and Figures 1 and 2 (CE-planes) as follows: Pages 19, 25, and 26: “PGI-I is presented as the difference between groups in the proportion of participants reporting improvement.”
 • Table 2: please explain why the standard deviation for the surgery intervention is zero, given that different types of POP surgery have different costs. 	Despite having other surgery options, all surgeries conducted were those with the same costs: 4640 Euros. This information has been added to the Results section as follows: Page 14: “Despite having other surgery options (Supplementary file 2), we used a fixed price of €4640 considering the surgical procedures conducted in the trial.”
 • Figures 1 and 2: the CE graphs would be helpful if the vertical cost axis contained zero, and lines show the zero axes, to clearly demark the quadrants of the cost-effectiveness plane. 	The CE-planes (Figures 1 and 2) have been updated to include the zero lines as shown below. Figures are now provided as Supplementary file 2. 

• Strengths and limitations section: text that consultations being done by gynecologists when usual care is for a primary care doctor: might be helpful to note that this is not necessarily true in health care systems of other countries.

As suggested, this has been added to the Discussion as follows:

Page 17: “Second, consultations related to both interventions were provided by gynecologists, which may result in an overestimation of intervention costs. This may not be representative for healthcare systems in other countries, as these consultations

	may be provided by trained GPs at lower costs (i.e., €39 by a GP vs €109 by a medical specialist).”
Comments of Reviewer #2	Response
A very minor point regarding this otherwise very clear report is that the likelihood of initial pessary treatment being cost-effective is 1 whereas the authors conclude 'pessary therapy is likely to be cost-effective compared to surgery', which is not in line. Why not boldly state 'initial pessary treatment is the preferred treatment based on the balance between costs and effects' (or something along these lines)? The fact that many do subsequently cross over is not a good reason to weaken this conclusion. This might be the 'step up' that would still save costs without losing on QoL etc.	Thank you for bringing this up. Nevertheless, as findings are based on models using a probabilistic approach, we think that keep the term likelihood would be more appropriate. We do see your point, though, and add your suggestion that pessary would save costs at an acceptable loss of effect as shown below: Page 15: “For QALYs, similar to PGI-I the positive, ICERs indicate that pessary therapy is less expensive and less effective (mean difference -0.01, 95% CI -0.05; 0.03) than surgery. However, the difference in QALYs was small and less than the commonly used minimally clinically important difference (i.e., 0.06)^{46,47} meaning that pessary therapy would save costs without considerably reducing health-related quality of life.” Page 16: “Main findings This economic evaluation showed that although non-inferiority of pessary therapy with regard to subjective improvement could not be shown which was consistent with primary analysis of PGI-I⁷. Also, there were no statistically significant differences in QALY gained. Despite this, a strategy of initial pessary therapy in women with symptomatic POP is likely to be cost-effective compared to immediate surgery from a healthcare and a societal perspective due to lower costs associated with pessary therapy.”
Comments of Reviewer #3	Response
The paper is well written and follows the CHEERS checklist. There is one issue that could be presented in a clearer way. The results of the CEAC are mainly in the SW quadrant. Although the benefit difference is not significant it is on average less for the pessary group. The authors should explain what this means in terms of the fact that it is not the willingness to pay for an additional QALY but more the willingness to accept the loss in benefit.	Thank you for pointing this out. Methods and Results Section has been updated to further explain the willingness to accept the loss in benefit. Pages 14, 15: “For the PGI-I outcome, the main analysis showed ICERs of 65525 and 67203 from a healthcare and a societal perspective, respectively (Table 3). The positive ICERs are situated in the SW quadrant of the CE plane and indicate that while pessary therapy incurred significantly lower costs (healthcare mean difference -€1780, 95% CI -€2148; -€1422 and societal mean difference -€1826, 95% CI -€2328; -€1322), it was also less effective compared to surgery (mean difference = -0.03, 95% CI -0.11; 0.06), although not statistically significantly so. Most bootstrapped cost-effect pairs were situated on the

	right of the non-inferiority margin for effects (95.5%) and in the southern quadrants of the CE-Plane meaning that pessary therapy would save costs at an acceptable loss of effect in terms of PGI-I (Figure 1[1A] and [2A]). Due to statistically significant lower healthcare and societal costs in the pessary therapy group compared to surgery, CEACs showed that the probability of the pessary therapy being cost-effective compared to surgery was 1 at relevant WTP values (Figure 1 [1B] and [2B]). This means that the pessary therapy as an initial treatment option has a 100% probability of being cost-effective compared to immediate surgery. < Insert Figure 1 here > For QALYs, similar to PGI-I the positive, ICERs indicate that pessary therapy is less expensive and less effective (mean difference -0.01, 95% CI -0.05; 0.03) than surgery. However, the difference in QALYs was small and less than the commonly used minimally clinically important difference (i.e., 0.06)^{46,47} meaning that pessary therapy would save costs without considerably reducing health-related quality of life. The majority of the bootstrapped cost-effect pairs was in the southern quadrants of the CE-plane (70%) meaning that on average the pessary was less costly than surgery (Figure 2 [1A] and [2A]). The probability that pessary therapy being cost-effective compared to surgery at all WTP thresholds was 1 from both perspectives (Figure 2 [1B] and [2B]).”
On page 24 Table 3 reports the different percentages of bootstrapped results in each of the different quadrants without explaining what the cost effectiveness plane is.	Thank you for pointing this out. To clarify, the Methods Section and Table 3 has been updated as suggested. Page 10: “Bootstrapped cost-effect pairs were described and plotted on cost-effectiveness planes (CE-planes)⁴¹.” Bottom of Table 3: “CE-plane = cost-effectiveness plane showing the difference in costs between pessary therapy and surgery on the y-axis and the difference in effects on the x-axis resulting in four quadrants namely, NE = northeast (pessary therapy more expensive and more effective than surgery); SE = southeast (pessary therapy less expensive and more effective than surgery); SW = southwest (pessary therapy less expensive and less effective than surgery); NW = northwest (pessary therapy more expensive and less effective than surgery).”

The paper could benefit from some discussion and interpretation of the ICERs as some are negative and the values are much higher than the willingness to pay threshold.

As suggestion ICERs have been further described and interpreted in the Results Section as follows:

Pages 14, 15: “For the PGI-I outcome, the main analysis showed ICERs of 65525 and 67203 from a healthcare and a societal perspective, respectively (Table 3). The positive ICERs are situated in the SW quadrant of the CE plane and indicate that while pessary therapy incurred significantly lower costs (healthcare mean difference -€1780, 95% CI -€2148; -€1422 and societal mean difference -€1826, 95% CI -€2328; -€1322), it was also less effective compared to surgery (mean difference = -0.03, 95% CI -0.11; 0.06), although not statistically significantly so. Most bootstrapped cost-effect pairs were situated on the right of the non-inferiority margin for effects (95.5%) (Figure 1[1A] and [2A]) and in the southern quadrants of the CE-Plane meaning that pessary therapy would save costs at an acceptable loss of effect in terms of PGI-I. Due to statistically significant lower healthcare and societal costs in the pessary therapy group compared to surgery, CEACs showed that the probability of the pessary therapy being cost-effective compared to surgery was 1 at relevant WTP values (Figure 1 [1B] and [2B]). This means that the pessary therapy as an initial treatment option has a 100% probability of being cost-effective compared to immediate surgery.

< Insert Figure 1 here >

For QALYs, similar to PGI-I the positive, ICERs indicate that pessary therapy is less expensive and less effective (mean difference -0.01, 95% CI -0.05; 0.03) than surgery. However, the difference in QALYs was small and less than the commonly used minimally clinically important difference (i.e., 0.06)^{46,47} meaning that pessary therapy would save costs without considerably reducing health-related quality of life. The majority of the bootstrapped cost-effect pairs was in the southern quadrants of the CE-plane (70%) meaning that on average the pessary was less costly (Figure 2 [1A] and [2A]). The probability that pessary therapy being cost-effective compared to surgery at all WTP thresholds was 1 from both perspectives (Figure 2 [1B] and [2B]).

On page 15 line 43/44 the authors report that pessary is cost effective at all WTP thresholds. Do they mean the thresholds they refer to? The figure values do not exceed the WTP threshold applied it would be worthwhile for the reader if higher values were included in the figures.

Willingness-to-pay values of the model-based economic evaluation of Hullfish KL, et al, ranged between \$0 to roughly \$20000 per QALY gained. In our study the range was similar, although in a different currency €0 to €20000 per QALY gained.

	The text has been updated to clarify that: Page 16: “They developed a model-based economic evaluation with 12-month follow-up based on data from the literature, local experience of a single institution, and expert opinion. Results showed that for lower WTP thresholds (i.e. from 0 to 5600 \$/QALY gained) pessary is cost-effective compared to surgery and for higher WTP thresholds (i.e., from 5600 to roughly 20000 \$/QALY gained) not anymore. Our results, based on randomized data, showed that pessary therapy is cost-effective compared to surgery at similar WTP thresholds (i.e. 0 to 20000 €/QALY gained).”												
The conclusions of the study should be placed in the context that the study is a short term study (24 months) and there is need for longer term follow-up as the prolapse symptoms can reappear and there is need for longer term follow-up studies.	As suggested, this has been included in the Discussion Section: Page 16,17: “Validated outcome measures were used and the trial had a follow-up of 2 years. However, since POP symptoms can relapse over time, studies including a longer follow-up (e.g., more than 2 years) are needed.”												
In the results section page 13 line 11 and line 21 there is reference to figure 1 did the authors mean supplementary figure 1 the consort diagram rather than the CEAC?	Yes, thank you. This has been corrected as follows: Page 13: “Of the 1605 women assessed for eligibility, 440 were randomized to either pessary therapy (n=218) or surgery (n=222) as shown in Supplementary file 2.”												
Authors should check Page table 1 line 9/10 the n values are the wrong way round.	These typos have been corrected, thank you. TABLE 1. BASELINE CHARACTERISTICS OF PARTICIPANTS    Baseline characteristic Pessary therapy n = 218 Surgery n = 221     Age (mean (SD)) 64.8 (9.5), n=218 64.7(9.2), n=221   Risk-increasing aspects [†] (n, %) 71 (32.6), n=218 221 58 (26.2), n=221 218   History of gynecological surgery (n, %) 22 (10.1), n=218 28 (12.7), n=221   	Baseline characteristic	Pessary therapy n = 218	Surgery n = 221	Age (mean (SD))	64.8 (9.5), n=218	64.7(9.2), n=221	Risk-increasing aspects [†] (n, %)	71 (32.6), n= 218 221	58 (26.2), n= 221 218	History of gynecological surgery (n, %)	22 (10.1), n=218	28 (12.7), n=221
Baseline characteristic	Pessary therapy n = 218	Surgery n = 221											
Age (mean (SD))	64.8 (9.5), n=218	64.7(9.2), n=221											
Risk-increasing aspects [†] (n, %)	71 (32.6), n= 218 221	58 (26.2), n= 221 218											
History of gynecological surgery (n, %)	22 (10.1), n=218	28 (12.7), n=221											
The CEACs are very hard to decipher. The point estimate is barely visible . The authors should consider including the WTP lines for more clarification as the vertical axes do not include 0 and positive values Also, they should consider incorporating all four quadrants of the cost effectiveness plane as they report them in the text.	Indeed without the zero reference line, CE-planes are hard to interpret. To incorporate the four quadrants of the CE-plane,the zero line in the y-axis and x-axis have been added to Figures 1 and 2(Supplementary file 2) as shown below:												

In the checklist the authors state that an HEAP was developed and reference page 5 but there is no mention of it in the study.	Thanks for pointing this out. The health economic analysis plan was part of the study protocol provided as supplementary material. To clarify that, a sentence has been added to the Methods section as follows: Page 5: “The health economic analysis plan is available in the study protocol provided as Supplementary file 1.”
There authors have not included a justification for the study horizon.	A justification has been added to the Methods section as follows: Page 7: “This economic evaluation was conducted from a healthcare and a societal perspective over a time horizon of 24 months based on the literature and as recommended by the National Institute for Health and Clinical Excellence^{6,8,18}.” References: van Geelen JM, Dwyer PL. Where to for pelvic organ prolapse treatment after the FDA pronouncements? a systematic review of the recent literature. Int Urogynecol J. 2013;24(5):707-718. doi:10.1007/s00192-012-2025-3 Bugge C, Adams EJ, Gopinath D, et al. Pessaries (mechanical devices) for managing pelvic organ prolapse in women. Cochrane Database Syst Rev. 2020;11(11):CD004010. doi:10.1002/14651858.CD004010.pub4

	National Institute for Health and Care Excellence. Urinary incontinence and pelvic organ prolapse in women: management. Published April 2, 2019. Accessed January 3, 2022. https://www.nice.org.uk/guidance/ng123
Although the authors include the cost year for the productivity costs there is no statement about all the other costs that are included in the study.	This information was included at the end of the paragraph describing the lost productivity costs, but it has now been moved up to the beginning of the paragraph describing the cost measures: Page 8: "All costs were indexed to 2022 using the consumer price index in the Netherlands(www.cbs.nl) ³¹ ."
The study does not include an economic model so questions 16,17.18 , 19 and 22 are not relevant to this study.	This is correct, thank you.

VERSION 2 – REVIEW

REVIEWER	Thomas, Sonia RTI International
REVIEW RETURNED	16-Jan-2024

GENERAL COMMENTS	The authors have responded to all comments and the manuscript is well written and nicely improved. Two remaining items related to the methods would benefit from further clarification: - the authors appear to have analyzed the change in PGI-I on the linear scale. Assuming this is true, it would be helpful to add "linear" at the top of page 10: "multilevel LINEAR regression models" - The authors added "PGI-I at 12-month and extra consultations due to complications, were included in the models as confounders as they were related to the outcomes and treatment allocation" yet this is not clear reasoning for why this would be done and how it impacts the analysis - could the authors provide further reasoning and a reference for why post-baseline outcomes would be included in a model of change from baseline.
--

VERSION 2 – AUTHOR RESPONSE

Comments of Reviewer #1

The authors have responded to all comments and the manuscript is well written and nicely improved.

Two remaining items related to the methods would benefit from further clarification:

- the authors appear to have analyzed the change in PGI-I on the linear scale. Assuming this is true, it would be helpful to add "linear" at the top of page 10: "multilevel LINEAR regression models"

Response to the Reviewers' comments

As suggested, the text has been updated to clarify that we analysed PGO-I on a linear scale as shown below:

Page 10: "Multilevel linear regression models were used to estimate the difference in costs and effects between the groups to account for the fact that randomization was stratified by hospital center[38]."

Comments of Reviewer #1

- The authors added "PGI-I at 12-month and extra consultations due to complications, were included in the models as confounders as they were related to the outcomes and treatment allocation" yet this is not clear reasoning for why this would be done and how it impacts the analysis - could the authors provide further reasoning and a reference for why post-baseline outcomes would be included in a model of change from baseline.

Response to the Reviewers' comments

We agree with the reviewer on this point. The models have been updated to exclude post-baseline outcomes. Results (including tables and figures) have been updated throughout the manuscript without affecting the main conclusions.

Page 10: "All analysis models were adjusted for relevant baseline confounders. The PGI-I model was adjusted for risk-increasing aspects and prolapse stage. The QALY model was adjusted for baseline utility values[39], risk-increasing aspects, and prolapse stage. Healthcare and societal costs models were adjusted for age, menopause state, risk-increasing aspects, and prolapse stage."

Page 12: "After adjusting for confounders, the lower 95% CI bound of the PGI-I outcome still surpassed the non-inferiority margin (mean difference -0.05, 95% CI, -0.14; 0.03, Table 3). There was no statistically significant difference in QALYs between groups neither in the unadjusted analysis (mean difference -0.02, 95% CI, -0.06; 0.02, Table 2) nor the adjusted analysis (mean difference -0.03, 95% CI -0.07; 0.002, Table 3).

Page 13: "For the PGI-I outcome, the main analysis showed ICERs of 33509 and 34295 from a healthcare and a societal perspective, respectively (Table 3). The positive ICERs are situated in the SW quadrant of the CE plane and indicate that while pessary therapy incurred significantly lower costs (healthcare mean difference -€1807, 95% CI -€2172; -€1446 and societal mean difference -€1850, 95% CI -€2349; -€1341), it was also less effective compared to surgery (mean difference = -0.05, 95% CI, -0.14; 0.03), although not statistically significantly so. Most bootstrapped cost-effect pairs were situated on the right of the non-inferiority margin for effects (83.2%) and in the southern quadrants of the CE-Plane meaning that pessary therapy would save costs at an acceptable loss of effect in terms of PGI-I (Figure 1[1A] and [2A])."

Pages 13 and 14: "For QALYs, similar to PGI-I the positive, ICERs indicate that pessary therapy is less expensive and less effective (mean difference -0.03, 95% CI -0.07; 0.002) than surgery. However, the difference in QALYs was small and less than the commonly used minimally clinically important difference (i.e., 0.06)[46,47] meaning that pessary therapy would save costs without considerably reducing health-related quality of life. The majority of the bootstrapped cost-effect pairs was in the southern quadrants of the CE-plane (100%) meaning that the pessary therapy was less costly than surgery (Figure 2 [1A] and [2A])."